# Polyamines in Plant–Pathogen Interactions: Roles in Defense Mechanisms and Pathogenicity with Applications in Fungicide Development

**DOI:** 10.3390/ijms252010927

**Published:** 2024-10-11

**Authors:** Qi Yi, Min-Jeong Park, Kieu Thi Xuan Vo, Jong-Seong Jeon

**Affiliations:** 1Graduate School of Green-Bio Science and Crop Biotech Institute, Kyung Hee University, Yongin 17104, Republic of Korea; daowomenglilai@khu.ac.kr; 2Department of Genetics and Biotechnology, Kyung Hee University, Yongin 17104, Republic of Korea; minj0331@khu.ac.kr

**Keywords:** polyamines, plant–pathogen interaction, fungicides, plant defense mechanisms, pathogenicity

## Abstract

Polyamines (PAs), which are aliphatic polycationic compounds with a low molecular weight, are found in all living organisms and play essential roles in plant–pathogen interactions. Putrescine, spermidine, and spermine, the most common PAs in nature, respond to and function differently in plants and pathogens during their interactions. While plants use certain PAs to enhance their immunity, pathogens exploit PAs to facilitate successful invasion. In this review, we compile recent studies on the roles of PAs in plant–pathogen interactions, providing a comprehensive overview of their roles in both plant defense and pathogen pathogenicity. A thorough understanding of the functions of PAs and conjugated PAs highlights their potential applications in fungicide development. The creation of new fungicides and compounds derived from PAs demonstrates their promising potential for further research and innovation in this field.

## 1. Introduction

The productivity of crops is influenced by both abiotic factors, such as drought, heat, and cold stress, and biotic factors, such as pathogens, insects, and herbivores. These factors can severely compromise plant health, deplete nutrients, and reduce yield. To survive, plants have developed various defense mechanisms, including physical barriers, chemical signals, and cellular responses. To combat pathogens, plants rely on two primary layers of immunity: pathogen-associated molecular pattern-triggered immunity (PTI) and effector-triggered immunity (ETI), which defend against a wide range of biotic stresses. PTI is activated by the recognition of common microbial signatures, while ETI provides a stronger response, initiated by specific pathogen effector proteins. During defense, plants depend on an array of metabolites, proteins, and signaling molecules that work in concert to detect and respond to pathogenic attacks. Together, these mechanisms enable plants to mount effective defenses and adapt to the ever-changing challenges in their environment.

Polyamines (PAs), positively charged aliphatic polycationic compounds found in all living organisms, play essential roles in growth and development [1,2,3,4]. In all organisms, PAs exist in two forms: free PAs or covalently conjugated PAs (CC-PAs). In plants, CC-PAs may account for up to 90% of total PAs in certain species [5]. Soluble CC-PAs, which constitute the largest pool of PAs in plants, are formed through covalent bonding with phenolic compounds, whereas insoluble CC-PAs are formed by electrostatic interactions with negatively charged macromolecules, such as nucleic acids, specific proteins like eukaryotic translation initiation factor 5A (eIF5A) [3,6,7,8,9]. PAs have been proven to participate in numerous cellular processes, including gene expression regulation, cell proliferation, the cell cycle, and the modulation of cell signaling. These activities contribute to important physiological events such as floral bud differentiation, flowering, pollen development, and seed development [10,11,12,13,14]. Notably, the role of PAs in abiotic stress has been extensively reviewed [7,15,16]. Recently, growing evidence has suggested the importance of PA biosynthesis in both plant defense and pathogen virulence during plant–pathogen interactions [16,17,18,19,20,21,22,23]. However, a complete understanding of the roles of PAs in biotic stress remains elusive. While pathogens utilize certain PAs to enhance their invasion capabilities, such as through sporulation and appressorium formation, plants harness specific PAs to bolster their immune defenses [24,25,26,27,28]. In this review, we compile recent studies on the roles of PAs in plant–pathogen interactions, offering a comprehensive review of their functions in plant defense and pathogen pathogenicity. Additionally, insights into PA functions have prompted their exploration in fungicide development, revealing significant potential for further research in crop improvement.

## 2. PA Metabolism

Putrescine (Put), spermidine (Spd), and spermine (Spm) are the most common PAs found in nature. Although their levels vary among organisms, Put and Spd are typically predominant, followed by Spm in plants and pathogens [1,29]. In plants, the homeostatic regulation of Spd and Spm is generally more effective than that of Put, resulting in wider fluctuations in Put levels, particularly under stress conditions [30].

There are four pathways to synthesize Put in plants, two of which require arginine, while the alternative pathways utilize ornithine and citrulline (Figure 1). Arginine is converted to agmatine by either arginine decarboxylase (ADC) or arginine amidohydrolase (ARGAH), both of which have been shown to function synergistically under stress conditions [4,31,32]. Agmatine is then converted into N-carbamoyl putrescine (NCP) and subsequently Put via two consecutive enzymatic steps, catalyzed by agmatine iminohydrolase (AIH) and the N-carbamoyl putrescine amidohydrolase (NCPAH). Alternatively, ornithine and citrulline are decarboxylated by ornithine decarboxylase (ODC) and citrulline decarboxylase (CDC), respectively, to produce Put [1]. The CDC pathway has, so far, been observed only in sesame (*Sesamum indicum*). In plants, the ADC and ODC pathways are well studied, with the ODC pathway linked to cell division, while the ADC pathway plays a role in stress responses [33,34]. The dominance of each pathway varies depending on cell type and stress conditions. However, the ODC pathway is absent in *Arabidopsis thaliana* and many members of the Brassicaceae family [35]. In bacteria, Put is synthesized via the arginine and ornithine pathways. Arginine is decarboxylated by ADC (also known as SpeA), followed by two steps catalyzed by AIH (also known as AguA) and NCPAH (also known as AguB) to form Put. Put can also be produced from ornithine via ODC (also known as SpeC) [29]. In most fungi, Put is synthesized exclusively through the ODC pathway (Figure 1) [6].

In all species, Put serves as a precursor for the synthesis of Spd and Spm. Specifically, Put is converted to Spd by spermidine synthase (SPDS) in plants, also known as SpeE in bacteria or SpdSyn in fungi, through the addition of an aminopropyl group. A second aminopropyl group is then added to Spd to form Spm via spermine synthase (SPMS) in plants, also known as Spe4 or SpmSyn in fungi [6,32,36] (Figure 1). Spm is relatively rare in bacteria and is not essential for the growth of the fungus *Saccharomyces cerevisiae* [6,29,36].

Diamine oxidase (DAO) and PA oxidase (PAO) are key enzymes involved in PA metabolism across plants and pathogens [1,37] (Figure 1). DAO catalyzes the conversion of Put into hydrogen peroxide (H_2_O_2_) and γ-aminobutyric acid (GABA), which serves as a precursor for succinate, feeding into the TCA cycle [6]. In fungi and bacteria, Put, Spd, and Spm are reversibly catalyzed by PAO, forming the intermediates N-acetylspermine and N-acetylspermidine [29,38]. In plants, however, PAOs catalyze the oxidation of Spd and Spm, producing H_2_O_2_ and releasing 1,3-diaminopropane, another PA substrate [33,34,39].

## 3. PA Transport

PAs are primarily synthesized in plastids and the cytosol in plant cells, while they are catabolized in the apoplast and peroxisomes [40,41]. The differing localization of PAs within cellular compartments underscores the importance of PA transport between these compartments and their roles in regulating various cellular processes. The transport of PAs is facilitated by a transmembrane electrical gradient and a potential antiport mechanism, first observed in carrot (*Daucus carota*) protoplasts. In these experiments, calcium concentration and the pH of the media solution were found to influence PA uptake into carrot protoplasts [42]. The first identified plant PA transporter, Spd-preferential OsPUT1, was discovered in rice (*Oryza sativa*). *OsPUT1* is expressed in all tissues except seeds and roots [41]. Additional rice transporters, OsPUT2 and OsPUT3, as well as Arabidopsis AtPUT1, AtPUT2, and AtPUT3, exhibit high substrate specificity for Spm uptake, as demonstrated when introduced into the yeast PA uptake mutant *agp2Δ* [43]. A transient expression assay in *Nicotiana benthamiana* revealed that OsPUT1 localizes to the endoplasmic reticulum, while AtPUT2 and OsPUT3 localize to the chloroplast [44]. In Arabidopsis protoplasts, transient expression showed that AtPUT1 (also known as AtLAT3) localizes to the endoplasmic reticulum, while AtPUT2 (also known as PAR1 or AtLAT4) localizes to the Golgi apparatus. Similarly, OsPUT3 (also known as OsPAR1) localizes to the Golgi apparatus in rice protoplasts [45]. In transgenic Arabidopsis plants, GFP-tagged AtPUT3 (also known as RMV1 or AtLAT1) localizes to the plasma membrane [46]. The overexpression of *RMV1* increased PA uptake activity compared to that in control lines. The differences in localization observed across these studies could be attributed to the use of different expression assays, cell types, or experimental conditions. However, their primary presence in transport-related organelles underscores their essential roles in cellular transport.

In tomato (*Solanum lycopersicum*), the Put2 and Put5 transporters regulate the import of Put and Spd, which further influences PA catabolism and enhances tolerance to salt stress [47]. The overexpression of *OsPUT1* or *OsPUT3* in *A. thaliana* delayed flowering time and resulted in higher levels of Spd and Spd conjugates in the leaves, suggesting that PA transporters play an important role in regulating flowering and senescence pathways [44]. However, the mechanism by which PA transporters respond to growth and environmental stimuli remains unclear.

In bacteria, PAs are transported by ATP-binding cassette (ABC) transporters. The PotABCD and PotFGHI systems are responsible for the import of Spd and Put, respectively, while the PotE and sapBCDF systems are involved in their secretion [29] (Figure 1). In the phytopathogen *Xanthomonas citri*, over 20% of ABC transporters are expressed upon infection, with PotFGHI being one of the most highly expressed transporters [48]. PotF, a periplasmic-binding protein in the PotFGHI system, is highly conserved in the *Xanthomonas* genus and exhibits a higher affinity for Spd than Put, consistent with Spd being twice the size of Put. Notably, tissues of *Citrus sinensis* infected with *X. citri* accumulated higher levels of Spd. In another example, the transcription of *PotA* in *Streptococcus pneumoniae*, a human pathogen, is induced under peroxide-induced oxidative stress but not under acidic stress, indicating that different environmental factors in various species modulate PA transporters [49].

In fungi, PA uptake is strongly associated with the non-transporting transceptor Agp2, which has lost the ability to transport ligands, as well as with the SAM transporter Sam3, the urea transporter DUR, and the SR protein kinase Sky1 [6] (Figure 1). In *S. cerevisiae*, four PA transporter (TPO) genes are involved in PA export, with TPO1 being the most relevant. *UGA4*, which encodes a γ-aminobutyric acid permease, has been identified as responsible for Put uptake in the vacuolar membrane of *S. cerevisiae* [50]. Additionally, the membrane transporter HOL1 is translationally repressed by PAs but strongly expressed under low PA conditions. HOL1 is essential for yeast growth under low-PA conditions, while SAM3, DUR3, and AGP2 are dispensable [51].

## 4. PA Roles in Plant–Pathogen Interactions

### 4.1. Endogenous PAs in Plant Immunity and Pathogen Pathogenicity

Accumulating evidence suggests that PA biosynthesis is essential for both plant defense and pathogen pathogenicity during plant–pathogen interactions (Figure 2). Upon pathogen infection, the levels of various PAs and activities of related enzymes change in both plants and pathogens, each with distinct roles [26]. In pathogens, PA accumulation enhances pathogenicity, particularly during the early stages of infection. The silencing of Spd synthesis in pathogens such as *Aspergillus flavus*, *Magnaporthe oryzae*, *Fusarium graminearum*, and *Alternaria alternata* results in a loss of growth and sporulation, defective appressorium formation, and the decreased activity of cell wall-degrading enzymes and mycotoxin production [25,52,53,54]. In plants, PAs are newly synthesized in infected tissues across various species, including maize (*Zea mays*) and *A. thaliana*, with Put levels notably increasing [17,19,55]. ADC, a crucial enzyme for Put synthesis, is often targeted in studies to modulate Put levels in plant–pathogen interactions. ADC-silenced lines show a significant reduction in PA levels and increased reactive oxygen species (ROS) content, which results in heightened pathogen susceptibility in *A. thaliana* [55,56,57,58]. Conversely, the overexpression of the *ADC* gene in *A. thaliana* and tobacco leads to increased Put levels and reduced in planta growth of *Pseudomonas syringae* [59,60]. A recent study revealed that foliar application of nano-selenium enhances melon defenses against fungal infections by boosting PA metabolism, revealing the antifungal properties of PAs [61].

Beyond their antifungal activity, PAs have also been implicated in plant immunity [26,62]. Recent studies indicate that PAs contribute to PTI and ETI in wheat (*Triticum aestivum*) and Arabidopsis [55,62,63]. The exogenous application of Put on Arabidopsis has been reported to induce callose deposition and increase the expression of several PTI maker genes [64]. Moreover, the interplay between PAs and ETI components has been observed. For example, the inoculation of *A. thaliana* with the *P. syringae* pathovar tomato (Pst) DC3000 carrying AvrRpm1 (Pst AvrRpm1) induces Put accumulation, whereas the R-gene *rpm1-1* mutant line shows significantly reduced Put accumulation, suggesting a role for Put in the RPM1 and AvrRpm1 interaction [55]. The effector CSEP087 from the grapevine powdery mildew *Erysiphe necator* isolate *NAFU1* interacts with arginine decarboxylase VviADC [65]. The transient overexpression of *VviADC* in grapevine leaves enhances resistance to powdery mildew, likely due to increased PA levels in infected tissues. AvrPiz-t interacting protein 5 (APIP5), a negative regulator of cell death, is targeted by the *M. oryzae* effector Avrpiz-t, which represses APIP5 transcriptional activity, leading to cell death [66]. *APIP5-RNAi* transgenic plants show enhanced resistance against *M. oryzae*. In rice, hydroxycinnamoyl Put, a phenolamide important for plant resistance to biotic stress, is synthesized from Put-hydroxycinnamoyl acyltransferase (*OsPHT4*) [67,68]. APIP5 directly represses the transcript level of *OsPHT4* by binding to A-box motifs in the *OsPHT4* promoter, regulating hydroxycinnamoyl Put synthesis. This suggests that hydroxycinnamoyl Put plays a role in the APIP5-mediated immune response associated with ETI (Figure 2). While some evidence shows that altered PA levels in host plants under infection trigger immune responses, there is still limited understanding of PA-regulating factors, and further investigation is needed to elucidate the detailed mechanisms involved.

### 4.2. Extracellular Secreted PAs in Plant Immunity and Pathogen Pathogenicity

Both plants and pathogens secrete PAs into the extracellular environment. *P. syringae* secretes Put into the host apoplast to promote virulence, suggesting its role in plant colonization [20,69] (Figure 2). Although the exact mechanism by which Put enhances *P. syringae* virulence is unclear, it is hypothesized that Put secretion may mitigate the effects of ROS produced by plant defense responses. Similarly, *Ralstonia solanacearum* secrets Put into plant xylem vessels, causing bacterial wilt disease [69].

Recent studies have shown that extracellular PAs can trigger an increase in endogenous PA levels within plants [70]. This could explain the observed rise in endogenous PA levels during infection, which coincides with the secretion of PAs by fungi into the plant extracellular space [69]. Beneficial rhizomicrobiota have been found to reduce plant stress in a similar way by secreting PAs, leading to increased fresh and dry plant weight, stimulating plant PA synthesis, suppressing pathogens, and regulating plant hormones, among other effects [71]. In plants, the levels of apoplastic Put and Spd increase following inoculation with *Pst* DC3000 in Arabidopsis [64]. Other plants, such as tobacco, tomato, and rice, have also been found to secrete PAs into the apoplast during defense responses, indicating that host PAs play a role in cell surface defense against pathogen infection.

In a different context, during phage infection in *Pseudomonas aeruginosa*, the injection of liner DNA from the phage to the bacterial cell triggers an increase in intracellular PA levels. This inhibits phage genome replication, possibly due to PAs binding to phage nucleic acids [70]. This finding suggests that secreted host PAs might similarly restrict pathogen replication or transcription by binding to a pathogen’s nucleic acids.

Interestingly, bacterial uptake of host-secreted Put enhances the production of plant cell wall-degradation enzymes and upregulates genes involved in chemotaxis and flagellar biogenesis, thereby influencing bacterial motility (Figure 2). In *Dickeya fangzhongdai* and *Dickeya zeae*, triple mutations in the PA transporter genes *PotF* and *PlaP*, as well as the PA biosynthesis gene *SpeA*, impair the transport of host Put into bacterial cells and reduce bacterial swimming motility and virulence [72,73]. During plant–pathogen interaction, both plants and pathogens secrete PAs at the host–pathogen interface. Plants secrete PAs as a defense mechanism, while pathogens release specific PAs to enhance their pathogenicity. However, the outcome is unpredictable, as both the plant and pathogen can exploit the secreted PAs to their own advantage.

## 5. Application of PAs in Fungicides

PAs and their natural derivatives are known for their various biological properties, including antifungal, antimicrobial, and antiviral activities [74,75,76]. Although Spm and Spd exhibit fungicidal activity, their potency is relatively limited compared to commercial fungicides like carbendazim [77]. As a result, several fungicides have been developed based on PA activity. For example, PA analogs such as AMXT-1505 have been shown to control *F. graminearum* [78], while AMXT-2455 and AMXT-3016 demonstrate promising results in reducing the viability of *A. alternata* [79]. PA-based fungicides are also being developed by targeting enzymes or substrates that block the PA metabolic pathway in fungi. Two common fungicides that utilize this strategy are α-difluoromethylornithine (DFMO) and guazatine, both of which inhibit key steps in PA metabolism (Figure 3).

### 5.1. DMFO

Put has been identified as the most potent inducer of deoxynivalenol (also known as vomitoxin), a naturally occurring mycotoxin produced by *F. graminearum* [80]. DFMO primarily targets ODC activity, thereby hindering the synthesis of Put and blocking mycotoxin production in pathogens (Figure 3). This is effective because ODC is the only Put synthesis pathway in most fungi, but it does not interfere with plant PA synthesis [81]. DFMO has been shown to reduce the growth of *A. alternata* and *F. graminearum,* as well as their mycotoxin production, and this inhibition cannot be fully restored by exogenous PA treatment [79]. However, fungi can circumvent DFMO inhibition by importing Put from the host during infection. This process can be blocked using Put transport inhibitors, such as AMXT-1505, which prevent the pathogen from acquiring host-derived Put [78].

### 5.2. Guazatine

Guazatine, a mixture of reaction products from PAs, is commonly used to protect cereals and fruits by inhibiting lipid biosynthesis and destabilizing membranes in fungi. However, guazatine also inhibits plant PAO, negatively impacting plant growth, even at micromolar concentrations (Figure 3) [82]. To mitigate this toxicity, genome-wide association mapping has been employed to investigate guazatine resistance-related genes in *A. thaliana* and *Geotrichum citri-aurantii*, the causative agent of a major postharvest disease in citrus. These studies have identified potential candidates for genetic editing to create guazatine-resistant plants [82,83]. Although guazatine and similar fungicides may not completely eliminate pathogens, the ability of PAs to reduce pathogen growth has led to ongoing research into the development of more PA-based fungicides.

The main challenge lies in understanding the precise mechanism by which PAs influence plant–pathogen interactions, as different types of PAs appear to function in unique ways. PAs have been observed to initially bind to the sugar–phosphate backbone of DNA and subsequently interact with the minor or major grooves of the DNA double helix. The stability of DNA-PA complexes increases as the positive charge of the PAs rises [84]. Recent studies on the effects of PAs on gene expression show that low concentrations of PAs reduce repulsion between negatively charged molecules, such as DNA and RNA, thereby promoting gene expression. In contrast, high concentrations of PAs lead to DNA compaction, resulting in the complete suppression of gene expression [85]. One key interaction involves Spd covalently binding with hypusine-containing eIF5A, which is found in all eukaryotes and is essential for cell proliferation [86]. PAs have been found to block eIF5A function in the termination of the peptide sequence MLLLPS∗ (∗ = stop codon) in the antizyme gene *HOL1* upstream open reading frame (uORF) or elongation in the polyproline motif of antizyme inhibitor 1 (AZIN1) in bacteria [51,87]. Similarly, in plants, inactive eIF5A leads to ribosome stalling, triggering the no-go decay pathway or inducing the non-stop decay pathway [88]. In all cases, higher levels of proteins with extended N-terminal regions are produced, starting from less optimal start codons. These findings have opened new potential avenues for the development of novel fungicides.

## 6. CC-PAs in Plants and New Strategies in Fungicide Development

The formation of CC-PAs has been revealed to be a critical step in controlling Put homeostasis within a non-toxic range for plant survival [59]. The increase in CC-PAs mitigates free PA levels, which would otherwise promote tumor growth in maize following *Ustilago maydis* infection [19]. The protective effect of exogenous PAs against damage from superoxide has been shown to depend on their prior conversion to conjugated forms [89]. Conjugates of PAs with hydroxycinnamic acids, such as coumaric acid, caffeic acid, and ferulic acid, collectively known as phenolamides, play a significant role in plant responses to biotic stress [90]. A dose–response relationship between the ratio of Put to Spd conjugates and the concentration of aflatoxin B1, the most toxic aflatoxin produced from *A. flavus*, demonstrates the stronger defense capability of Put conjugates against aflatoxin contamination in maize seeds [91]. Hydroxycinnamic acid amides, a subclass of phenolamides, have been widely studied for their role in enhancing plant resistance by inhibiting spore germination on leaf surfaces, stimulating plant immune responses including the jasmonic acid, salicylic acid, and abscisic acid pathways, and boosting the production of lignin, callose, and ROS [92].

A recent study discovered that a new salt composed of a plant resistance inducer and a PA cation could potentially mitigate the effects of both abiotic and biotic stresses. Tobacco plants treated with this new salt showed fewer necrotic spots following tobacco mosaic virus infection compared to those treated with unmodified inducers [93]. Additionally, star polycations have been developed as gene or pesticide nanocarriers, offering low cytotoxicity and high delivery efficiency [94]. Star-PAs were used in an avermectin B1a (AVM) nano-delivery system to disrupt AVM self-aggregation. By breaking down the self-aggregated AVM structures, star-PAs facilitated more uniform dispersion at the nanoscale, enhancing systemic transport in plants and increasing both contact and stomach toxicity to pests [95]. This highlights the importance of PAs in improving agricultural practices.

Beyond pathogen control, PA-based fungicides could also contribute to crop production and improved ecotoxicological safety [96]. For instance, pyraclostrobin, a widely used fungicide, is highly toxic to aquatic organisms. However, pyraclostrobin-loaded polyurea microcapsules, fabricated with PAs, have been shown to control peanut leaf spots, increase peanut yields, and reduce toxicity to zebrafish [96]. Further research is needed to explore the effectiveness of this approach in other crops. Moreover, postharvest treatment with Put has been shown to extend the shelf life and enhance the quality of apricots [97], suggesting that new PA-based fungicides could improve both pathogen defense and crop quality.

The growing demand for PAs and their analogs in agriculture reflects their recognized importance. However, their complex natural structures and limited availability present challenges for conventional engineering methods. Recent advancements in metabolic engineering have enabled the large-scale production of PAs and PA-containing natural derivatives through the manipulation of *S. cerevisiae* metabolism pathways [76]. This breakthrough increases the potential applications of PAs in fields ranging from agriculture to material science.

## 7. Conclusions

Put, Spd, and Spm are the most common PAs in nature, and they play distinct roles in both plants and pathogens during their interactions. Advances in our understanding of PA synthesis, secretion, and function have led to the development of PA-based fungicides aimed at inhibiting PA metabolism in pathogens. Two commonly used PA-based fungicides target different steps of PA metabolism: (1) DFMO targets the PA synthesis enzyme ODC, the sole pathway for PA synthesis in fungi. However, its efficacy can be compromised if pathogens import Put from the host. (2) Guazatine disturbs membrane function and inhibits PAO function in plants, which may negatively affect plant health.

Given the limitations of current PA-based fungicides, new biotechnological approaches involving CC-PAs offer promising alternatives. Novel fungicides that combine plant resistance inducers with PA cations or employ nano-delivery systems using star-PAs with AVM represent exciting strategies for improved pathogen control. Additionally, the development of ecotoxicologically safe fungicides that enhance crop production and extend storage life is another critical direction of future research.

In summary, while PAs play a pivotal role in plant–pathogen interactions, the strategic manipulation of these compounds through advanced biotechnological innovations holds great potential for enhancing plant resistance, creating more effective fungicides, and improving crop production and storage.

## Figures and Tables

**Figure 1 ijms-25-10927-f001:**
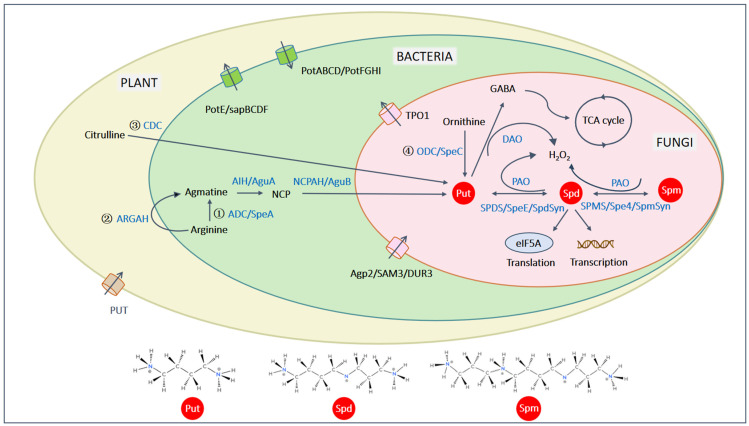
The Venn diagram illustrates the distinct and overlapping pathways in PA metabolism across plants, bacteria, and fungi. Put is synthesized through four pathways: (1) From arginine via ADC/SpeA, AIH/AguA, and NCPAH/AguB, a pathway common to plants and bacteria. (2) A plant-specific pathway involves ARGAH, which works with ADC under stress conditions. (3) A pathway found only in sesame converts citrulline to Put via CDC. (4) A universal pathway that begins with ornithine and involves ODC/SpeC present in plants, bacteria, and fungi. Put is converted into Spd by SPDS/SpeE/SpdSyn through the addition of an aminopropyl group, while Spm is produced from Spd by SPMS/Spe4/SpmSyn through the addition of another aminopropyl group. DAO degrades Put into H_2_O_2_ and GABA. In plants, PAO facilitates the conversion of Spd and Spm, including Spm to Spd, and Spd to Put, with H_2_O_2_ as a byproduct. The chemical structures of Put, Spd, and Spm are illustrated. All enzymes are represented in blue. Put, putrescine; Spd, spermidine; Spm, spermine; NCP, N-carbamoylputreseine; ADC/SpeA, arginine decarboxylase; AIH, agmatine iminohydrolase; AguA agmatine deiminase; NCPAH/AguB, N-carbamoylputreseine amidohydrolase; ODC/SpeC, ornithine decarboxylase; CDC, citrulline decarboxylase; ARGAH, arginine amidinohydrolyase; SPDS/SpeE/SpdSyn, spermidine synthase; SPMS/Spe4/SpmSyn, spermine synthase; DAO, diamine oxidase; PAO, polyamine oxidase; GABA, γ-aminobutyric acid.

**Figure 2 ijms-25-10927-f002:**
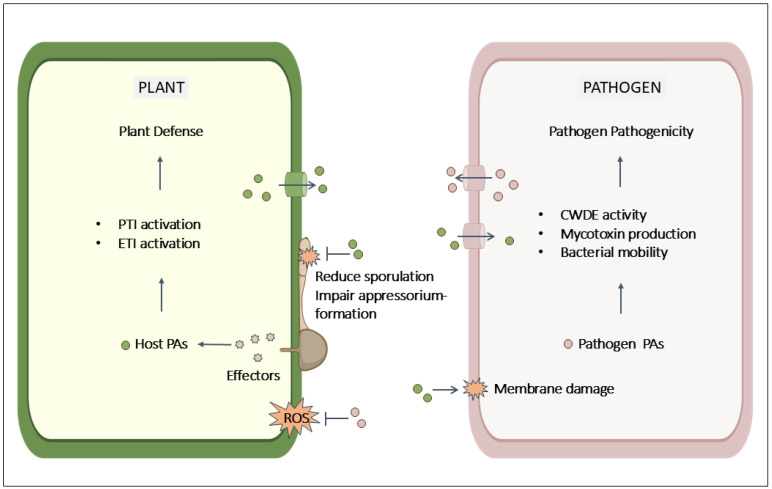
Proposed model illustrating the role of PAs in plant immunity and pathogen pathogenicity. During plant–pathogen interactions, PAs produced by plants activate the host immune response, while PAs produced by pathogens increase the activity of cell wall-degrading enzymes (CWDEs), boost mycotoxin production, and enhance bacterial motility. Host-derived PAs are secreted to inhibit the growth of bacteria and fungi by damaging microbial membranes, decreasing sporulation, and disrupting appressorium formation. Conversely, PAs secreted by pathogens reduce plant ROS levels, facilitating pathogen colonization. Additionally, pathogens can utilize host-derived PAs to enhance their own pathogenic capabilities. Green and pink globules represent plant-synthesized PAs and pathogen-derived PAs, respectively. ROS, reactive oxygen species. This figure was created using BioRender (https://biorender.com/ (accessed on 30 September 2024)).

**Figure 3 ijms-25-10927-f003:**
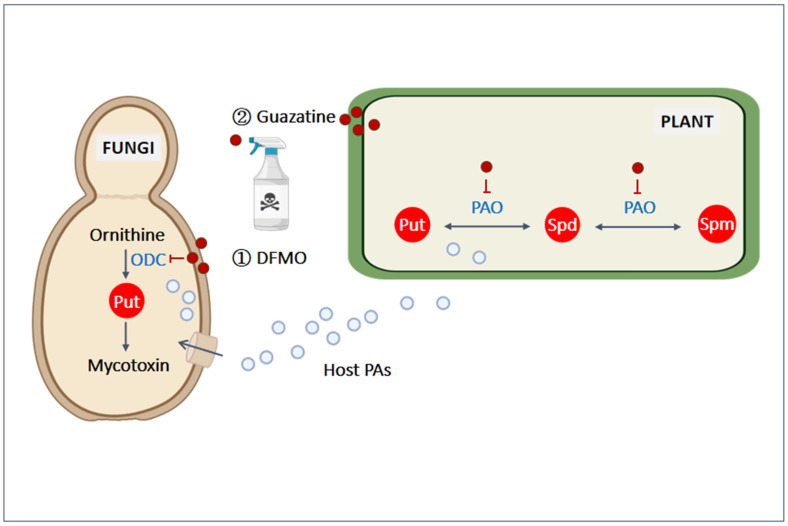
The mechanism of action of two commonly used PA-based fungicides. (1) DFMO inhibits the enzyme ODC, which is the sole pathway for PA synthesis in fungi. (2) Guazatine disrupts membrane function and reduces cellular permeability in fungi but may also inhibit PAO activity in plants, potentially negatively impacting plant health. DFMO, α-difluoromethylornithine; ODC, ornithine decarboxylase; PAO, polyamine oxidase. Red globules represent fungicides, while light blue globules represent plant-synthesized PAs. This figure was created using BioRender (https://biorender.com/ (accessed on 30 September 2024)).

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
