# Peer review of "Polyamines in Plant–Pathogen Interactions: Roles in Defense Mechanisms and Pathogenicity with Applications in Fungicide Development"

_ijms, 2024, doi:10.3390/ijms252010927_

Round 1

Reviewer 1 Report

Comments and Suggestions for Authors

In this manuscript, the authors summarized published results on the role of polyamines in plant-pathogen interactions:.

They showed up-to-date results on the plant defense mechanisms induced by polyamines.

The role of PA on plant stress tolerance is well known and therefore, this article is just important addition to known facts.

However, the authors focused “Fungicide Development” in the title but no relevant information is provided in the text and not even any dedicated section.

The authors mostly discussed basic facts about PA like Polyamines Biosynthesis, Catabolism, and Transport; role in agriculture etc. rather that mechanistic approaches and underlying mechanism toward the development of fungicides through molecular approaches.

Many of the information is superficial and like a textbook.

Future perspectives are not relevant to the discussed facts.

Author Response

Dear Reviewer,

We would like to express our gratitude to you for your valuable feedback. In response, we have thoroughly reviewed recent literature and made substantial revisions in line with your suggestions. The entire manuscript has been restructured, and updated versions of three figures have been included. We have also incorporated recent findings on polyamine action at the molecular level and highlighted promising strategies for the development of polyamine-based fungicides for enhancing crop protection. Major changes are highlighted in yellow.

Comment 1: In this manuscript, the authors summarized published results on the role of polyamines in plant-pathogen interactions. They showed up-to-date results on the plant defense mechanisms induced by polyamines. The role of PA on plant stress tolerance is well known and therefore, this article is just important addition to known facts. However, the authors focused “Fungicide Development” in the title but no relevant information is provided in the text and not even any dedicated section. The authors mostly discussed basic facts about PA like Polyamines Biosynthesis, Catabolism, and Transport; role in agriculture etc. rather that mechanistic approaches and underlying mechanism toward the development of fungicides through molecular approaches.

Response 1: Thank you for highlighting these important points. From the plant’s perspective, endogenous PAs are synthesized during infection to regulate immunity and are secreted into the apoplast to defend against pathogen attacks. Conversely, in pathogens, endogenous PAs promote growth, while secreted PAs can suppress host defense mechanisms. One of the key remaining questions is how each specific type of PA contributes to host defense and pathogen virulence. Several mechanisms have been proposed to explain the effects of PAs, including their involvement in ROS production and the antimicrobial activity of conjugated PAs (CC-PAs). Additionally, recent findings have shown that PAs can covalently bind to nucleic acids and eukaryotic translation initiation factor 5A, shedding light on this complex mechanism. In the revised version of the manuscript, we have made substantial improvements to address this comment. Notably, we revised sections 4.1. Endogenous PAs in plant immunity and pathogen pathogenicity, 4.2. Extracellular secreted PAs in plant immunity and pathogen pathogenicity, 5. Application of PAs in fungicides, and 6. CC-PAs in plants and new strategies in fungicide development. In section "6. CC-PAs in plants and new strategies in fungicide development," we have compiled current knowledge on conjugated PAs and explored their potential applications in fungicide development. In addition, we reviewed recent studies demonstrating the effectiveness of conjugated PAs with salts in enhancing plant defense. Furthermore, we discussed in detail the potential role of PAs in the development of ecotoxicologically safe fungicides.

Comments 2: Much of the information is superficial and like a textbook. Future perspectives are not relevant to the discussed facts.

Response 2:  We greatly appreciate your careful review and thoughtful suggestions. Polyamines have long been recognized as key players in abiotic stress tolerance and, more recently, in plant-pathogen interactions. While there have been several reviews on the role of PAs in plant-pathogen interactions, most have focused on how PAs contribute to plant defense or pathogen virulence without exploring the precise mechanisms, largely due to limited understanding. We acknowledge that this is a relatively new area of research, and the lack of advanced techniques has constrained more in-depth studies. In this revised version, in addition to summarizing the roles of PAs in plant-pathogen interactions, we have incorporated recent findings on the molecular mechanisms of PA action as described in Response 1 and offered new insights into their potential applications in crop protection. 

Reviewer 2 Report

Comments and Suggestions for Authors

In the manuscript "Polyamines in Plant-Pathogen Interactions: Roles in Defense Mechanisms and Pathogenicity with Applications in Fungicide Development”, Yi et al compile recent studies on the roles of polyamines (PAs) in plant defense and pathogen pathogenicity. The authors provided brief summary for PAs biosynthesis, catabolism, and transport, and roles of PAs in plant hosts and pathogen pathogenicity. Although this is an important topic in biochemistry and molecular biology, the authors did not review and summarize the latest articles for readers. Other new review articles in this topic are published and constructed much better than this. Detailed data or summary should be listed as a table for comparison.

Lines 179, Erysiphe necator NAFU1, bigger word size for?

Roles of PAs in pathogen pathogenicity should be made as in table or figure.

New papers especially published in the latest 5 years including review articles should be reviewed thoroughly. Overall, this manuscript is not ready for publication and this reviewer recommends unacceptance for publication in ijms.

Author Response

Dear Reviewer,

We would like to express our gratitude to you for your valuable feedback. In response, we have thoroughly reviewed recent literature and made substantial revisions in line with your suggestions. The entire manuscript has been restructured, and updated versions of three figures have been included. We have also incorporated recent findings on polyamine action at the molecular level and highlighted promising strategies for the development of polyamine-based fungicides for enhancing crop protection. Major changes are highlighted in yellow.

Comment 1: In the manuscript "Polyamines in Plant-Pathogen Interactions: Roles in Defense Mechanisms and Pathogenicity with Applications in Fungicide Development”, Yi et al compile recent studies on the roles of polyamines (PAs) in plant defense and pathogen pathogenicity. The authors provided brief summary for PAs biosynthesis, catabolism, and transport, and roles of PAs in plant hosts and pathogen pathogenicity. Although this is an important topic in biochemistry and molecular biology, the authors did not review and summarize the latest articles for readers. Other new review articles in this topic are published and constructed much better than this. Detailed data or summary should be listed as a table for comparison.

Response 1: Thank you for your insightful review and suggestions. PAs are known for their role in abiotic stress tolerance and, more recently, in plant-pathogen interactions. Although some reviews discuss their role in plant defense and pathogen virulence, few have addressed the detailed mechanisms due to limited knowledge. We acknowledge that this field is still emerging, and the lack of advanced techniques has hindered further exploration. In this revised version, in addition to summarizing the roles of PAs in plant-pathogen interactions, we have included recent discoveries on the molecular mechanisms of PA activity and provided new insights into their potential applications in crop protection. A key remaining question is how each specific type of PA contributes to host defense and pathogen virulence. Several mechanisms have been proposed, including the involvement of PAs in ROS production and the antimicrobial activity of conjugated PAs (CC-PAs). Additionally, recent findings have shown that PAs can covalently bind to nucleic acids and eukaryotic translation initiation factor 5A, shedding light on this complex mechanism. To address this comment, we have made substantial improvements in the revised manuscript, particularly, in sections 4.1. Endogenous PAs in plant immunity and pathogen pathogenicity, 4.2. Extracellular secreted PAs in plant immunity and pathogen pathogenicity, 5. Application of PAs in fungicides, and 6. CC-PAs in plants and new strategies in fungicide development.

Comment 2: Lines 179, Erysiphe necator NAFU1, bigger word size for?

Response 2: NAFU1 is an isolate of Erysiphe necator collected from a severely infected Vitis vinifera cv. Rizamat plant in a vineyard in Northwest China during the summer of 2011 (https://doi.org/10.1016/j.plaphy.2015.11.003). We have revised the text to reflect this, editing the part to “Erysiphe necator isolate NAFU1”. Thank you.

Comment 3: Roles of PAs in pathogen pathogenicity should be made as in table or figure.

Response 3: Thanks for your comments. We have revised Figure 2 to enhance the visualization of PA contributions, making the relationships and mechanisms clearer.

Comment 4: New papers especially published in the latest 5 years including review articles should be reviewed thoroughly. Overall, this manuscript is not ready for publication and this reviewer recommends unacceptance for publication in ijms.

Response 4: In response to your comments, we have reviewed the latest literature on PAs. While numerous studies focus on abiotic stress, relatively few address plant-pathogen interactions, and most of these emphasize how PAs contribute to plant defense or pathogen virulence without delving into the specific mechanisms involved. In this revised version, we have cited and highlighted the key points from these reviews while incorporating recent findings on the molecular actions of PAs. These include the involvement of PAs in ROS production and the antimicrobial activity of conjugated PAs (CC-PAs). Additionally, recent studies have shown that PAs can covalently bind to nucleic acids and eukaryotic translation initiation factor 5A, shedding light on this complex mechanism. Furthermore, we have included information on conjugated PAs and their potential applications in fungicide development, demonstrating their effectiveness in crop protection.

Reviewer 3 Report

Comments and Suggestions for Authors

Dear authors,

I consider that you must draw all chemical structures you cite in the manuscript in order to reach up major audience. Furthermore, I wonder what is the new vision you provide in comparison with other recent reviews based on the same topic.

Best regards

Comments on the Quality of English Language

Fine

Author Response

Dear Reviewer,

We would like to express our gratitude to you for your valuable feedback. In response, we have thoroughly reviewed recent literature and made substantial revisions in line with your suggestions. The entire manuscript has been restructured, and updated versions of three figures have been included. We have also incorporated recent findings on polyamine action at the molecular level and highlighted promising strategies for the development of polyamine-based fungicides for enhancing crop protection. Major changes are highlighted in yellow.

Comment : I consider that you must draw all chemical structures you cite in the manuscript in order to reach up major audience. Furthermore, I wonder what is the new vision you provide in comparison with other recent reviews based on the same topic.

Response 1: Thank you for bringing this to our attention. We have added PA structures to Figure 1 to improve visualization. PAs are well known for their roles in abiotic stress tolerance, and numerous studies have focused on this aspect. More recently, however, PAs have been recognized for their involvement in biotic stress. While some reviews discuss PAs in plant-pathogen interactions, most focus on their contribution to plant defense or pathogen virulence, without exploring the underlying mechanisms, largely due to limited understanding. In this revised version, we have introduced several new components, particularly, in sections 4.1. Endogenous PAs in plant immunity and pathogen pathogenicity, 4.2. Extracellular secreted PAs in plant immunity and pathogen pathogenicity, 5. Application of PAs in fungicides, and 6. CC-PAs in plants and new strategies in fungicide development. Notably, Figure 1 illustrates the biosynthesis of PAs in plants, bacteria, and fungi, highlighting both the shared and unique pathways in each group. This sets the foundation for discussing fungicides that target PA biosynthesis in fungi in the subsequent section. In addition to summarizing the contributions of PAs in both plants and pathogens, we have compiled recent findings on their molecular actions. Importantly, we have included information on conjugated PAs and their potential applications in fungicide development, demonstrating their effectiveness in enhancing crop protection.

Round 2

Reviewer 1 Report

Comments and Suggestions for Authors

The current version is acceptable.

Reviewer 2 Report

Comments and Suggestions for Authors

The manuscript has been improved.

Reviewer 3 Report

Comments and Suggestions for Authors

Dear authors,

Thanks for your reply.

Comments on the Quality of English Language

Fine